# Perceptions of virtual primary care physicians: A focus group study of medical and data science graduate students

**Courtney M. Goetz** *, **Judith E. Arnetz**, **Sukhesh Sudan**, **Bengt B. Arnetz**

Department of Family Medicine, College of Human Medicine, Michigan State University, East Lansing, Michigan, United States of America

* goetzcou@msu.edu

## Abstract

### Background

Artificial and virtual technologies in healthcare have advanced rapidly, and healthcare systems have been adapting care accordingly. An intriguing new development is the virtual physician, which can diagnose and treat patients independently.

### Methods and findings

This qualitative study of advanced degree students aimed to assess their perceptions of using a virtual primary care physician as a patient. Four focus groups were held: first year medical students, fourth year medical students, first year engineering/data science graduate students, and fourth year engineering/data science graduate students. The focus groups were audiotaped, transcribed verbatim, and content analyses of the transcripts was performed using a data-driven inductive approach. Themes identified concerned advantages, disadvantages, and the future of virtual primary care physicians. Within those main categories, 13 themes emerged and 31 sub-themes.

### Discussion

While participants appreciated that a virtual primary care physician would be convenient, efficient, and cost-effective, they also expressed concern about data privacy and the potential for misdiagnosis. To garner trust from its potential users, future virtual primary physicians should be programmed with a sufficient amount of trustworthy data and have a high level of transparency and accountability for patients.

## Introduction

As artificial intelligence (AI) technology continues to advance, healthcare systems are adapting care delivery to incorporate these technologies and enhance the services they provide [1–3]. Moreover, the ongoing COVID-19 pandemic has dramatically accelerated the use of telehealth and AI-informed decision-making in healthcare [4–6]. Additionally, patients have begun to

**Data Availability Statement:** All de-identified transcripts have been attached as Supporting Information.

**Funding:** This study was funded by Dr. Bengt Arnetz's Michigan State University start-up funds.

**Competing interests:** The authors have declared that no competing interests exist.

rely on artificial intelligence to inform their healthcare decisions. For example, with the introduction of online tools such as symptom checkers, healthcare consumers utilize primary care differently, often reducing their urgency to receive care or intended level of care (urgent care, primary care, hospital, etc.) [7]. Patients have also begun to utilize other healthcare technologies, such as telemedicine, receiving primary care via text or video chat [8]. Further, young people seem to be using healthcare differently than previous generations, generally choosing more convenient and cost-effective options for their care [9].

The incorporation of artificial intelligence and other technologies into everyday healthcare has elicited both confidence and suspicion from providers and healthcare teams. Those who are concerned about the use of AI in healthcare commonly cite the potential for the technology to replace healthcare workers, although there is little empirical evidence to support this [10]. Other concerns include issues of accuracy, accountability, transparency, and privacy [10]. There is also a dearth of empirical evidence of the benefits of AI on patients and consumers [2]. A particularly interesting development in AI is virtual primary care. In contrast to other at-home telehealth healthcare options such as phone and video visits, virtual primary care would allow patients to receive care virtually from an AI "physician." Insurers, such as Humana, continue to expand their offerings of healthcare plans centered around virtual care. Employers like Amazon and Walmart are also starting to offer virtual care to their employees [11]. Companies such as Babylon Health have already developed AI doctors with similar diagnostic accuracy to human physicians [12]. The market for AI in healthcare has been projected to grow to $27 billion a year in 2025 [13]. Despite this, there is little peer-reviewed research on the impact of switching to virtual care on providers or patients.

## Virtual primary care

*Virtual primary care*, an artificial intelligence system—not a human physician- used via audio/videoconferencing, chat, or email, is becoming more possible as technology advances [1,10,12]. It is therefore not unrealistic to imagine that a virtual primary care physician may be a reality in the not-too-distant future [1,12,14]. Technology is likely to transform healthcare, and considerable developments have already been made. One impactful development in health technology is *remote monitoring*, which is the use of digital technologies to monitor and capture health data from patients that is electronically transmitted to healthcare systems. This technology has utility for conditions like diabetes and hypertension, where data like blood glucose and blood pressure can be automatically monitored by AI systems [15,16]. Recently, technology has also markedly contributed to the field of radiology. For example, artificial intelligence has recently been shown to be more accurate than radiologists in breast cancer prediction [17], and to have similar or better accuracy than dermatologists in diagnosis of melanoma [18,19].

While it is clear that artificial intelligence technologies may transform healthcare, less is known about the perspectives of potential users. In fact, a review study concluded that, rather than developing additional algorithms and systems, researchers should be seeking the perspectives of patients and consumers to ensure that the technology will actually be used by patients, and that it will have a meaningful impact on their health [2]. As younger generations age and begin to make their own healthcare decisions, their opinions are especially relevant. Data from the 2018 U.S. Census show that young adults ages 19–34 are the most likely to be uninsured [20]. In addition, a recent Blue Cross Blue Shield Association study showed that one in three millennials do not have a primary care physician, instead choosing more convenient and cost-effective options like urgent care and retail clinics [9]. Technology use is a potential strategy to increase healthcare access and utilization among young people, perhaps while reducing costs

[1,21]. Research shows that younger generations use more modern technology [22] and are more likely than older generations to be early adopters of new technologies [23]. For example, various studies of direct-to-consumer telehealth utilization demonstrate that a majority of patients are young adults [21,24,25]. It has also been demonstrated that those who are uninsured are likely to prefer telehealth over a traditional office visit [25], and that telehealth may increase utilization among patients who otherwise would not have sought care [21].

In order to better understand the perspectives of younger adults, the current study explored the attitudes of current medical/technology-focused graduate students toward virtual primary care physicians (vPCPs), artificial intelligence systems that diagnose and provide care. We chose to focus on advanced degree students in these disciplines because they were likely to have an informed understanding of healthcare or engineering, which might enhance their understanding of this new technology. In addition, research has shown that higher education level may predict early adoption of health technology [26,27]. Of note, this study took place in early 2019, before the COVID-19 pandemic. As such, these results should be considered in context.

## Methods

### Participants

Participants were recruited from two midwestern universities to participate in focus groups. Mass recruitment emails were sent to students in medical school and data science graduate programs. Five first-year medical students and three fourth-year medical students were recruited, along with four first-year engineering graduate students and three fourth-year computer/data science graduate students. We choose these groups to represent those who had a working knowledge of care provision, as well as those who had an understanding of the science behind AI technology. In addition, we chose students in the first and fourth (final) year of their schooling, in order to assess possible differences based on level of education and experience. The fourth-year students were also likely to have more experience using healthcare as a patient, as a factor of age. While this study was determined exempt by the Institutional Review Board of the investigators' university, all participants signed a consent form explaining that the sessions would be recorded, but their responses would not be linked back to them as individuals. Participants were provided with a meal during the focus groups and were also sent a $25 gift card for their participation.

### Data collection

Data collection took place between March and June in 2019. One-hour focus groups were held with each of the four participant groups, i.e., first-year medical students, fourth-year medical students, first-year engineering/data science graduate students, and fourth-year graduate students. One researcher facilitated the focus groups, while two others were note-takers. The same researcher (JA), who has expertise in conducting focus groups, facilitated each group. Each focus group was audiotaped. Researchers developed a list of questions that were aimed to elicit students' opinions about their willingness to use, as a patient, a virtual primary care provider (vPCP), the pros and cons related to use, and how they imagined it would be operationalized (S1 Appendix). These questions were developed by the research group in a series of discussions informed by relevant literature and past experience conducting focus group research. When conducting the focus groups, the facilitator followed a question script in order to limit variation between groups. We defined virtual primary care physician as "an artificial intelligence system, not a physician via audio/videoconferencing, chat, or email". In addition, we iterated to the medical students that we wanted to hear their opinions of using a vPCP as a

patient, rather than incorporating a vPCP into their future clinical practice. Upon completion of the fourth focus group, no new themes emerged and it was determined that thematic saturation was achieved. Thus, no further focus groups were held [28].

## Data analysis

The audiotaped focus group discussions were transcribed verbatim. Qualitative content analysis of the transcribed texts was conducted using a data-driven inductive approach to code content into themes [29].

To begin, two researchers, one who was present during the focus groups and another who was not, examined transcripts of each focus group separately, coding dominant responses. These codes were then aggregated into main themes. The two coders compared these themes jointly and discussed agreements and differences. The initial agreement between the coders was about 80%. These coders then reevaluated the data until agreement was reached and no new themes emerged. A third researcher who was present during the focus groups then read the four transcripts and validated these findings.

We hypothesized that the technology-focused graduate students would be more supportive of the vPCP than the medical students. We assumed that the graduate students would have a more thorough knowledge of artificial intelligence technology, and thus would not feel as much uncertainty as the medical students regarding virtual care. In addition, we imagined that the medical students might be more concerned than the graduate students about the loss of human connection in their care. We also hypothesized that, compared to the fourth-year students, the younger first-year students in both disciplines would be more supportive of using a vPCP. We imagined that the younger students would be more likely to consider convenience and cost when making decisions about their medical care, while the older students might place more value on the human connection, since they were likely to have more experience as a patient.

## Qualitative rigor

The consolidated criteria for reporting qualitative research (COREQ) were used to guide data collection and reporting [30]. Qualitative rigor was fulfilled using Guba and Lincoln's criteria (creditability, transferability, dependability, and confirmability) as a guide [31]. Credibility (analogous to internal validity) was achieved by comprehensiveness in data collection and analysis. All coders became thoroughly familiar with the data by reading through the transcripts multiple times. Transferability (analogous to external validity) was assured by using verbatim quotes as relevant examples given by students from multiple ages and academic disciplines. Dependability (analogous to reliability) was achieved by using one coder who was not involved in the data collection. Confirmability was achieved through triangulation, involving three researchers, one of whom had not been present during the focus group discussions. All coders analyzed the verbatim reports, then validated findings amongst themselves.

## Findings

The analysis revealed that distinct themes emerged that were readily categorized according to the focus group discussion question areas of advantages, disadvantages, and the future of vPCPs. Within those main categories, 13 themes emerged and 31 sub-themes. An overview of the main themes and sub-themes is provided in Table 1.

**Table 1. Themes and subthemes of vPCP focus groups.**

| | Theme | Subthemes | Definition |
|---|---|---|---|
| **Advantages** | **Convenience/access**: More convenient and easier to access than a human physician | Routine illnesses | Convenient to use for common illnesses like common cold, upper respiratory infections, etc. |
| | | Transportation | Do not have to travel to clinic/hospital |
| | | Anytime, anywhere | Can use vPCP outside of business hours, during weekends, from any location |
| | **Efficiency**: More efficient than human physician | One-stop shop | Can address multiple concerns, multiple specialties during one visit |
| | | Knowledge support | Can provide additional information and second opinions to physician |
| | | Physician support | vPCP can provide basic support (documentation, simple visits, etc.) so that physicians can concentrate on more complex tasks (diagnostics, complex patient care, etc.) |
| | **Lower cost**: Would cost less than a human physician, for patients and systems | Money | Less expensive for systems, potentially less expensive for patients |
| | | Time | Saves patient time for travel, waiting, etc |
| | | Manpower | Less front desk/support staffing may be needed; reduced demands on at-home caregivers |
| | **Accuracy**: More accurate than human physician | Human error | Lower likelihood of mistakes |
| | | Information capacity | Humans are limited in their capacity to remember information, machines can hold nearly unlimited information |
| | | Diagnostic bias | May be more likely to consider all possibilities, physicians may be biased to common illnesses |
| | **Reduced stigma**: A vPCP would not operate using stigma or bias | Discrimination | Cannot discriminate by race/ethnicity/socioeconomic status, etc., consciously or unconsciously |
| | | Embarrassment | Can speak about "embarrassing" issues more freely, lack of human contact can reduce stigma/embarrassment |
| **Disadvantages** | **Data security**: Patient medical data could be stolen or hacked | Information theft | Information could be susceptible to theft, hacking, unauthorized sharing |
| | **Humanness**: Human interaction would be lost when using a vPCP | Physical exam | Cannot perform physical exam |
| | | Shared decision-making | Patients aren't involved in the decision-making process |
| | | Patient compliance | Compliance may be lower if the patient does not receive instructions from a human physician |
| | **Misdiagnosis**: How would mistakes be handled when using a vPCP? | Accountability | Who is accountable when mistakes happen? Company, hospital, data source? |
| | | Reporting inaccuracy | Mistakes could be made if patients misunderstand their symptoms or enter misleading information |
| | **Suitability**: A vPCP would not be appropriate in all cases | Rare conditions | A vPCP could not make decisions about rare or new conditions, since there is not enough data available |
| | | Mental health | Human connection is important for those experiencing depression, social isolation, etc. |
| **Future** | **Physicians**: a vPCP would be used in tandem with a human physician | Cannot replace | Cannot ever replace a human physician. Can only be a tool for them. |
| | | Data checking | Human physicians must check and verify the data entered into the vPCP |
| | **Patients**: A vPCP could be used to enhance patient care | Patient engagement | Patient education and engagement could be personalized instead of generic printouts generated by the electronic health record |
| | | Chronic conditions | Easier to monitor chronic conditions |
| | **Population health**: A PCP would be a valuable epidemiological tool | Epidemiology | Could monitor disease prevalence and outbreaks, locally and worldwide |
| | **Data technology**: A future vPCP must include certain technologies | Dynamic system | System must change and add new data constantly as we learn new things |
| | | Adequate testing | Must be sufficiently tested and proven before rollout |
| | | Trustworthy data | Data must be sufficient and informed by multiple sources without interests |
| | | Transparency | Algorithm source (design/funding), data ownership, sharing/selling of data must be transparent to patients |

## Advantages

Students' perceptions of the advantages of vPCPs could be categorized into five main themes: convenience/access, efficiency, standardized care, low cost, and accuracy (Table 1). A total of 15 sub-themes emerged, with multiple subthemes within each main theme.

**Convenience/Access.** Participants felt that a vPCP would be convenient to use and would increase access to primary care. This theme could be separated into three subthemes: routine illnesses, transportation, and anytime/anywhere.

Students felt that a virtual PCP could be used for routine illnesses, for example:

". . . cold, flu, or high-blood pressure. . . These very common diseases, I think [a] virtual physician can handle. . ." (First year medical students)

Students noted that using a vPCP would eliminate transportation demands for the patient:

"I was ill last semester. . . And it would have been really helpful for me not to have to drive to a clinic. . ." (First year medical student)

Participants felt it was advantageous that the vPCP could be accessed anytime and anywhere, including in rural areas and underdeveloped countries where access to healthcare is limited.

". . ..you can basically contact whenever you have the problem.. you can access it really easily." (First year graduate student)

**Efficiency.** Students agreed that using a vPCP would make some healthcare processes more efficient, both for patients and providers. This theme could be further defined into three subthemes: one-stop shop, knowledge support, and physician support.

The students viewed the vPCP as a sort of one-stop shop for all specialties:

"If you use a computer, I would think. . . it should be a specialist on everything." (First year graduate student)

Participants noted that the vPCP could provide knowledge support to a human physician.

"If I was in the room, and essentially got two doctors, basically, cause you got the machine and the doc, that would be another cool thing." (Fourth year medical student)

The participants believed that the vPCP could complete simpler tasks (documentation, prescription orders, etc.) so that human physicians could focus on and complete more complex tasks.

"I could imagine a physical could be something the machine could do 'cause it's not really something you need to diagnose. Just putting in a bunch of data." (Fourth year medical student)

**Lower cost.** Participants believed that using a virtual PCP would bring about lower costs for both patients and systems, directly and indirectly. Three major themes emerged: money, time, and manpower.

Students felt that using a virtual PCP would cost less than visiting a human physician.

"Cheaper and faster. . .. as a student, I really care about like, how much I would pay for a visit in a hospital sometime." (Fourth year graduate student)

They noted that using a virtual PCP would result in less time being spent in waiting rooms or traveling to clinics or hospitals, saving time.

"I don't wanna waste the time to go wait in the waiting room. . . whereas I could just sit home in my pajamas and talk to someone for 5 minutes–in and out." (First year medical student)

The use of a vPCP would also lessen manpower demands, both for clinics and at-home caregivers.

". . .there are caregiver roles that you have to play for family members, and if this work could be done by the tool, it could actually be very helpful." (First year medical student)

**Accuracy.**    Increased accuracy was perceived as being an advantage of using a vPCP. From this theme, three subthemes emerged: human error, information capacity, and diagnostic bias.
The groups felt that the vPCP would not be inhibited by human error or emotional decision making, leading to a lower likelihood of mistakes.

"I don't see any involvement of, like, emotion at that point, which is like something they tell to doctors. Like, 'Keep your emotions away,' right?" (Fourth year graduate student)

Students also acknowledged that a virtual PCP has a much greater capacity for information than a human physician.

"[H]uman brains are not really fully used. . . They cannot memorize everything. . . you cannot remember every single patient. . . what do they look like, what's their disease look like, whatever. They may make mistakes. . . AI can record all the information." (Fourth year graduate student)

The students added that a vPCP may have less diagnostic bias. They visualized the ideal vPCP as incorporating large amounts of data from patients worldwide, potentially eliminating diagnostic bias towards locally frequent illnesses:

". . .it might catch some of the more obscure diagnoses or things that are frequently missed from a normal human physician perspective." (Fourth year medical student)

**Reduced stigma.**    The students felt that using a vPCP might result in reduced stigma. This theme could be dissected into two subgroups: discrimination and embarrassment.
Students felt that using a vPCP may reduce bias or discrimination. A few students gave personal stories of instances where they have felt stigmatized while using medical care.

". . . there are certain stigma or certain assumptions people make based on how they look. . . You don't have to deal with that when you're dealing with a tool." (First year medical student)

While this was a recurring theme, it was not shared among all of the students. A few students thought that availability of a vPCP could lead to increased discrimination related to

medical care. For example, one student expressed concern that the vPCP would be made available only to poorer or uninsured patients, while wealthier patients would be able to see a human physician.

The groups thought that the lack of human contact might make it easier for patients to speak about stigmatized or "embarrassing" things, such as sexually transmitted infections and mental health issues.

"And in many cases, you won't feel like sharing what you have, what you're going through with another person. So, in that case, for maintaining your confidentially or secrecy. It would be better to be with the machine. . .maybe if you have a machine you can talk, you can tell what you're going through." (First year graduate student)

### Disadvantages

The students in each of the focus groups saw a number of disadvantages with using a virtual PCP, relating to the following themes: data security, humanness, misdiagnosis, and suitability (Table 1).

**Data security.** This theme had one subtheme: information theft. Each group was concerned about the potential for their personal information to be stolen, sold, or otherwise shared without their consent:

". . .it could be sort of sold, it could be hacked, it could be taken into a wrong direction." (Fourth year medical student)

"I would never consult a virtual physician. . . The data. . . would go into hands of someone that I do not trust." (First year graduate student)

**Humanness.** Another disadvantage that was discussed in our focus groups was the lack of humanness that would come with using a vPCP. From this theme came three subthemes: physical exam, shared decision-making, and patient compliance.

There was concern among the groups about the lack of a physical exam when using a vPCP:

". . .the complete lack of a physical exam. I wouldn't be examined. My heart wouldn't be listened to, my lungs wouldn't be listened to.. the complete lack of touch, the complete[ly] lack of a physical exam would bother me." (First year medical student)

Another concern was that the vPCP would not involve the patient in the decision-making process.

". . .having that shared discussion as far as these are the positives and negatives of this treatment. . .that removes a lot of the personal decision that come into healthcare." (Fourth year medical student)

The students also believed that patient compliance would be lower when using a vPCP.

"[I]f it were to write prescriptions, maybe I wouldn't take them, who knows. Like, it wouldn't be as serious to me as a real physician. . ." (First year graduate student)

*Misdiagnosis.* Another major concern was the potential for misdiagnosis. Two subthemes emerged: accountability and reporting inaccuracy.

The students questioned who would be responsible for mistakes made by the vPCP. Many students expressed that this would need to be made clear to them before they would use a virtual physician.

"...it's worse if a computer makes the mistake because then the idea is, well, who do you sue? Whose fault is it?" (Fourth year medical student)

They also theorized that the vPCP could be easily misled by a patient's misunderstanding of their own symptoms, as their reporting to the vPCP may not always be clear and straight forward:

"So, if I'm not aware of what I'm going through, I might [put something] wrong into that, I might enter that I'm dealing with some other disease." (First year graduate student)

**Suitability.**  Students in each group agreed that a vPCP would not be suitable in all cases. Two themes emerged from these discussions: rare conditions and mental health.
Students believed a vPCP would not be appropriate for rare conditions:

"What if an event that has never been present, that was never used to train that AI system, presents itself in the future? What happens?" (Fourth year graduate student)

Some students believed a vPCP would be inappropriate for mental health concerns, especially in conditions like depression where human connection is important:

"...so much of medicine is not medicine. It's being a person, being a listener, being somebody you can talk to..." (Fourth year medical student)

## Future of vPCP

Students were asked to discuss how they envisioned a vPCP would work in the future. Multiple themes emerged, centered around physicians, patients, population health, and data technology (Table 1).
**Physicians.**  Much of the discussion around the future of virtual primary care centered around how it might affect human physicians. From these discussions, two subthemes emerged: cannot replace and data checking.
Students agreed that the vPCP would be a tool for the human physician, but could not replace them:

"Not necessarily as a stand-in for a physician. I just see it as a way of having 10,000 brains about this one problem as opposed to just one." (First year medical student)

The students also noted that any data or algorithm used by the vPCP would need to be checked and verified by human physicians.

"You end up building some kind of algorithm and have it take all these factors in consideration, then you test it... you have human physicians check it... anytime the computer makes a diagnosis or prescribes a treatment or whatever, then a physician would look at what the computer's doing and say, okay. Yes." (Fourth year medical student)

**Patients.**  Another theme centered around how a future vPCP might function to benefit patients. Two themes emerged: patient engagement and chronic conditions.

Students saw the potential for more personalized patient engagement when using the vPCP.

". . .it would be nice if the AI came in and took exactly what the doctor said and was able to formulate something specific to you, you know, not generic, not just the standard machine jargon, specifically." (Fourth year medical student)

They also envisioned that the vPCP would be beneficial for continuous monitoring of chronic conditions, for example:

"You could have someone who's a diabetic, and you feed the artificial intelligence data about your last blood glucoses and your A1C's without leaving your home." (First year medical student)

**Population health.**   A theme that arose in multiple groups was the use of a vPCP for population health and epidemiology.

"[If] it was like a virus, and it actually, it was like spreading, and so the person who was, like, helping, like, nurses and all, they're also getting infected. . .If there would have been a virtual machine to do that. . . let's put the machine inside the room; let's not enter there. . .. I feel like some extent, like, in some areas, we definitely need AI." (First year graduate student)

"[I]n addition to that, epidemiologically. . . we could predict real-time health outbreaks that are occurring. . . And all of a sudden, we have epidemiologic data that suggests, oh, in this region the outbreak is concentrated. It was this batch of lettuce that was contaminated. We can trace it back, and so I see a place for that." (First year medical student)

**Data technology.**   Students also discussed how the data technology might function in a future vPCP. Four subthemes emerged: dynamic systems, adequate testing, trustworthy data, and transparency.
Students discussed the importance of a dynamic system.

. . . (if) the system is very open source and, like, doctors and stuff can keep putting information in there, constantly, then it might be a very robust virtual physician." (First year graduate student)

They also iterated that any vPCP must be adequately tested before its release:

"So, if I know that it's an effective tool, it's been proven, it's been vetted, that would make me feel more comfortable." (First year medical student)

Students expressed that data entered into the vPCP must come from or be verified by multiple trustworthy sources without competing interests.

"[Y]ou have this human checking, was this product right or wrong. . . we have to make sure that those companies. . .they're not implementing a product just for the sake of having it tomorrow. . .we need some professionals in the field, and everybody who is trying to implement this sort of AI would have to meet [their] requirements before they can actually send it to hospitals" (Fourth year graduate student)

Finally, the students urged transparency regarding the data source used by the vPCP. They would want to know who was developing and implementing the system, the data that was being used to train the vPCP, how the vPCP was tested during development, and how their data would be used:

"Knowing who designed it and whether it was a physician or a group of physicians or a hospital. . . what dataset did they use. I'd want to know more about that." (First year medical student)

### Differences

The researchers also analyzed differences between groups. No major differences were identified between focus group participants at different education levels (first vs. fourth year). Between academic disciplines, opinions differed regarding two themes: mental health and trust. Medical students expressed hesitancy over using a vPCP for mental health concerns:

"Well, I think difficult topics, difficult situations, more psych-related issues, things that you actually just want to talk to a person about, you lose the relatability. 'Cause a lot of times, even docs, they are very relatable people." (Fourth year medical student)

Conversely, students in the engineering/data science groups thought that a vPCP would be easier to talk to:

"Like, as a human, there is some point where you really don't want to express everything to a human. . .without the fear of like, what does the other one think about you?" (Fourth-year graduate student)

While medical students generally did not trust vPCPs, citing data privacy and potential for misdiagnosis, the engineering/data science students were supportive.

". . .we certainly live in an era of big data, but like, I think that I would not be okay with sharing my information. . . I don't even care if it's depersonalized. I don't care if it's numbers attached without my name. I wouldn't want to share my information with anybody." (First year med student)

". . .when they would come out, they will be at the same levels or better than human. And as time goes by, they will only get better." (Fourth-year grad student)

## Discussion

While there has been extensive work on the algorithms and data science behind a virtual physician, and some companies have already developed virtual physicians, [12], research on the perceptions of its potential users has been limited [2]. Accordingly, this study aimed to understand how young people think about using a vPCP. We hypothesized that the technology-focused graduate students would be more supportive of the vPCP than medical students. This hypothesis was supported. Graduate students expressed that they would feel comfortable seeking care from a vPCP, while medical students were less trusting, citing data privacy concerns. We also hypothesized that the younger first-year students would be more supportive of using a vPCP than the fourth-year students, as a function of age and education experience. This hypothesis was not supported, as no differences were found between these groups.

Young people are already using health care differently than previous generations and are likely to be early adopters of this virtual primary care technology [9,23]. Appropriately, convenience and cost were two of the major advantages discussed during the focus groups. Another major advantage was efficiency. The students conceptualized the vPCP as being a helping tool for a human physician, something that could fill in on "simpler tasks" including documentation, treating common illnesses, and patient education. They theorized that this would free up the physician for more complex tasks, such as diagnostics, complex patient care, and even research and medication development. These ideas map closely onto the prominent concerns of current physicians, who face high productivity and administrative demands, contributing partly to elevated levels of burnout [32] and moral distress [33].

Students debated whether a vPCP would alleviate or exacerbate existing care disparities and stigmatization in healthcare. Many students visualized the vPCP as a way to avoid potential stigmatization, since a vPCP would not have any unconscious biases. However, some students thought that the vPCP may exacerbate inequalities and care disparities. Considering the substantial existing disparities in insurance status and access to healthcare in the US [34], these concerns were reflective of existing literature, suggesting they would likely need to be addressed in order for a vPCP to be widely accepted [35].

Students felt that a vPCP would have increased diagnostic accuracy over a human physician in many cases. Artificial intelligence systems have already demonstrated similar or higher diagnostic accuracy to human physicians, in primary care [12] and other disciplines [14]. Conversely, though, a major concern was the potential for misdiagnosis. In order to be accurate, an AI-physician system would need to have a sufficient amount of reliable, trustworthy, and validated data input from humans. The technology would also need to be sufficiently developed to understand and interpret the communicated input from human patients. Moreover, vPCPs are limited only to the information that has been entered by humans, and thus cannot know things that humans have not discovered and input to the machine. Accordingly, students wondered what might happen in these instances. They felt it was unclear who they might contact regarding mistakes and other concerns–would it be the company who developed the system, the hospital whose data is used, or someone else?.

Students also expressed some major concerns regarding data privacy and accountability for mistakes. This made sense, considering information hacking and privacy rights are major concerns in broader society today. Privacy issues are a common concern about AI in healthcare broadly [10]. The students cited examples of large-scale information theft and unauthorized information sharing, and for those reasons, some students were hesitant about sharing their health information with an artificial intelligence system. Many students noted they would not automatically trust a vPCP, and that transparency would be required in order to garner their trust.

Another perceived disadvantage to many of the participants was the lack of human connection that would come with using a vPCP. Many students saw this as an integral piece of their healthcare experience, especially in respect to mental health. The loss of humanity in medicine is also a concern of many who feel wary of artificial intelligence in healthcare [35]. However, some of the data science/engineering students didn't share this concern, and thought patients would eventually adapt, just as humans have adapted to other technologies like calculators, cell phones, and personal computers. Regardless of these concerns, all groups thought that a vPCP could be a valuable tool when working in tandem with a human physician, rather than replacing them. This echoes existing literature about the role of artificial intelligence in healthcare [1,3,35].

## Implications

These results offer insight into the potential development and implementation of virtual primary care systems. Some of the advantages, such as increased efficiency [1] and accuracy [3], have already been proposed in the literature as benefits of using AI in healthcare. However, a few of the major concerns, such as data safety, accountability, and health equity, haven't been proposed in research as patient concerns. Likewise, primary health care providers and health informatics experts shave been shown to hold similar concerns about AI systems in patient care [36]. Healthcare systems, AI developers, and policymakers should take these concerns into consideration as virtual care develops. Organizations should consider the perceptions of these consumers and make the necessary adjustment to their systems. For example, transparent and easy-to-understand privacy policies should be distributed to the user before their first use. All data used by the virtual PCP should be sufficient for reliable diagnoses, checked and verified by multiple disinterested parties, and these conditions should be explained to all patients using the system.

The students also discussed how virtual physicians have the potential to either exacerbate or alleviate existing care disparities, considering both the AI system itself and the context in which it is used. While it could be an affordable option for those who are not covered by health insurance, it is important that vPCPs do not replace in-person care for these populations, while more privileged patients are free to choose their method of care. In addition, data should not be used or shared in a way that can disadvantage certain groups of people, such as to make decisions about eligibility for health insurance coverage [37]. While the vPCP may expand access-to-care, there is still potential for bias.

In addition, care must be taken during the development of AI physician systems to ensure that data from a range of gender identities and racial and ethnic backgrounds is included [37]. The impact of relying on data from non-diverse datasets has already been demonstrated in the field of medical research. For example, a 2016 study [38] showed that only about 5% of genetic variants associated with asthma in European Americans were replicated in African Americans [39]. This means that evidence-based treatment strategies for asthma, which have been developed in research using mostly white European-Americans, are likely not the best fit for African-Americans. Correspondingly, deaths due to asthma in the U.S. are nearly 10 times higher in non-White children and over 2 times higher in non-White adults [40]. Moreover, there have already been recognized instances of racial bias in healthcare AI. One study found that a major risk-prediction algorithm assigned lower risk scores to Black patients than to White patients with comparable health conditions, attributable to mistakes in the algorithm's initial design [41]. There are many more examples of the impact of disparities in research [39,42,43], and AI [44,45] and it is crucial to not repeat this oversight when developing a virtual physician.

According to the focus group participants, setting standards for transparency and accountability would be necessary before they would use a virtual physician. This should start early with the development of the virtual physician, and continue throughout its implementation and use so as to reduce and prevent associated harms [37,46]. One potential barrier to complete transparency is known as the "black box" of AI, wherein decision-making processes used by AI are so complex that even engineering cannot decipher them [37,46]. Virtual physicians should be designed so that diagnoses and other decisions can be explained [46]. These explanations could be provided to patients at request, upon which they could share them with a human physician for a second opinion [46]. If it is not possible for decisions to be explained, extreme care should be taken during development of the AI to reduce the risk for errors [37]. The diagnostic sensitivity and specificity of the vPCP should also be explicitly published and

providers -and patients- should be educated about its accuracy [37,46]. A reporting structure should be established between implementers of the vPCP and its developers, and implementers should be expected to report instances of social bias or poor decision-making [37,46]. Of course, these are just a fraction of the ethical issues that should be considered before implementation of virtual physicians.

Considering the ongoing pandemic of COVID-19, these results are even more important. Interestingly, the potential for virtual primary care to be used in epidemiology and disease control was a theme that emerged in the Future of vPCP category, discussed in several of the focus groups. One student noted that the vPCP would be advantageous in active infection disease situations, as patients could receive care without the risk of healthcare workers becoming infected. Now, many primary care clinics have shifted quickly to telemedicine, and AI-based screening tools for COVID-19 are accessible on the Internet [47,48]. People worldwide may have to practice social distancing to some extent for months to come [49]. In addition, high infection rates are causing high demand for physicians [50]. Virtual primary care is possibly more relevant now than ever. Patient perceptions like those in this study should be used to optimize vPCPs for future use. Patients will ultimately decide whether or not to use these systems, so their opinions and concerns should be addressed during the development and implementation of virtual primary care.

## Strengths and limitations

To the best of our knowledge, this is the first study to explore young adults' perceptions of using vPCPs for health care. Specifically, this study explored perceptions of using a virtual primary care physician among advanced degree students. Since these groups had an informed understanding of either healthcare or engineering, they provided a good starting point to inform future research about patient perceptions of AI in healthcare and virtual physicians. Considering that these groups are already potential users and will soon enter the workforce as potential developers of vPCPs, their opinions are especially interesting. However, it may also be considered a limitation of the study that participants were young adults in graduate programs. Although they represented different disciplines, their perceptions may not be generally representative of young adults of a similar age, especially those with limited access to higher education. Future research should seek perceptions from patients of all ages and educational backgrounds.

## Supporting information

**S1 Appendix. Focus group questions.**
(DOCX)

**S1 File.**
(ZIP)

## Acknowledgments

The authors would like to thank the focus group participants for their participation.

## Author Contributions

**Conceptualization:** Courtney M. Goetz, Judith E. Arnetz, Sukhesh Sudan, Bengt B. Arnetz.

**Data curation:** Courtney M. Goetz, Judith E. Arnetz, Bengt B. Arnetz.

**Formal analysis:** Courtney M. Goetz, Judith E. Arnetz, Sukhesh Sudan.

**Funding acquisition:** Bengt B. Arnetz.

**Investigation:** Courtney M. Goetz, Bengt B. Arnetz.

**Methodology:** Courtney M. Goetz, Judith E. Arnetz, Bengt B. Arnetz.

**Project administration:** Courtney M. Goetz.

**Resources:** Bengt B. Arnetz.

**Supervision:** Judith E. Arnetz, Bengt B. Arnetz.

**Validation:** Courtney M. Goetz, Judith E. Arnetz.

**Visualization:** Courtney M. Goetz, Judith E. Arnetz, Bengt B. Arnetz.

**Writing – original draft:** Courtney M. Goetz.

**Writing – review & editing:** Courtney M. Goetz, Judith E. Arnetz, Sukhesh Sudan, Bengt B. Arnetz.

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
