## [Decision Letter · Decision Letter 0]

15 Sep 2020

PONE-D-20-10736

Perceptions of virtual primary care physicians: a focus group study of medical and data science graduate students

PLOS ONE

Dear Dr. Goetz,

Thank you for submitting your manuscript to PLOS ONE. After careful consideration, we feel that it has merit but does not fully meet PLOS ONE’s publication criteria as it currently stands. Therefore, we invite you to submit a revised version of the manuscript that addresses the points raised during the review process.

We look forward to receiving your revised manuscript.

Kind regards,

Maria Rosaria Gualano, MD, MPH

Academic Editor

PLOS ONE

Journal Requirements:

2. When reporting the results of qualitative research, we suggest consulting the COREQ guidelines: http://intqhc.oxfordjournals.org/content/19/6/349. In this case, please consider including more information on the number of interviewers, their training and characteristics; how participants were recruited; and whether bias issues were considered.

Reviewers' comments:

Reviewer's Responses to Questions

**Comments to the Author**

1. Is the manuscript technically sound, and do the data support the conclusions?

Reviewer #1: Partly

2. Has the statistical analysis been performed appropriately and rigorously? 

Reviewer #1: N/A

3. Have the authors made all data underlying the findings in their manuscript fully available?

Reviewer #1: Yes

4. Is the manuscript presented in an intelligible fashion and written in standard English?

Reviewer #1: Yes

5. Review Comments to the Author

Reviewer #1: The paper address an interesting topic, made even more urgent in light of the recent pandemic. Nevertheless some major concerns must be raised. In particular considering the subjects involved in the focus groups. In fact, students and graduate with medical and technological background were enrolled in the study. Firstly, it is quite hard to believe this population can be considered representative of young adults. Additionally, you stated that "we iterated to the medical students that we wanted to hear their opinions of being a patient of a virtual PCP, rather than their opinions from a provider perspective". I personally believe that this is quite unrealistic. In fact, the qualitative approach used to address the topic is based on subjective point of view and it sounds hard to believe that the subjects involved were able to split their opinion as patient from the one as provider.

Some more considerations:

Introduction:

You present references talking about Artificial Intelligence and telemedicine. These are completely different digital tools, please clarify and focus on the specific topic.

Line 36: Please, provide a reference to support this statement

Line 37: Please, provide a reference to support this statement

Line 53: Please, provide a definition of Virtual Primary Care.

Line 54-55: "While it is clear that technology may transform healthcare, less is known about the perspectives of its potential users". Please discuss more in depth this interesting issue (how healthcare can be transformed buy technology?) and provide more references.

Lines 58-61: This point should be discussed more in depth, particularly focusing on how technology can improve this condition.

Methods:

- How the subjects were recruited? They were selected randomly from the student list? Was the project presented in the schools and interested subjects contacted the researcher? Please provide further information on this crucial issue.

- "One researcher facilitated the focus groups, while two others were note-takers". The researcher was specifically trained to facilitate focus group? Some techniques were used to avoid the researcher to influence the respondents?

- The facilitating researcher was the same in the different focus group? If no, how ensure uniformity?

- How the questions were developed?

Discussion:

The discussion section sounds a repetition of the results section. Please provide more in depth discussion of the results.

6. PLOS authors have the option to publish the peer review history of their article (what does this mean?). If published, this will include your full peer review and any attached files.

Reviewer #1: No

---

## [Author Response · Author response to Decision Letter 0]

12 Oct 2020

Response to reviewer comments

Manuscript number: PONE-D-20-10736

“Perceptions of virtual primary care physicians: a focus group study of medical and data science graduate students”

Response: Our thanks to the reviewer for constructive comments and suggestions that have helped us to improve our paper. We provide responses in italics to each point below.

Comments from academic editor:

 Response: The title page has been reformatted according to these requirements.

2. When reporting the results of qualitative research, we suggest consulting the COREQ guidelines: http://intqhc.oxfordjournals.org/content/19/6/349. In this case, please consider including more information on the number of interviewers, their training and characteristics; how participants were recruited; and whether bias issues were considered.

 Response: These details have been incorporated into the methods section of the manuscript.

Response: We will provide de-identified versions of our focus group transcripts as Supporting Information files. We will specify this change in our cover letter.

Reviewers' comments:

Reviewer #1: The paper address an interesting topic, made even more urgent in light of the recent pandemic. Nevertheless some major concerns must be raised. In particular considering the subjects involved in the focus groups. In fact, students and graduate with medical and technological background were enrolled in the study. Firstly, it is quite hard to believe this population can be considered representative of young adults. 

Response: The reviewer raises an interesting and important point. Generalizability in qualitative research is a topic of debate [e.g.,1-3]. Qualitative research often seeks to explore perceptions or experiences that might be attributed to larger groups but where the main objective is to gain a better understanding of meaning or phenomena – not necessarily to generalize [1]. Rather than being representative of all young adults, we chose groups whose perceptions likely reflect the next generation of healthcare and data science professionals. These groups are likely to be early adopters of new health technology, and their background knowledge was beneficial for understanding this new technology. They are also likely to have some knowledge about what is available and possible regarding health technology. We explain this in more detail in the methods section. We also recognize that this is a limitation of our study, since our sample is likely not representative of all young adults, and we have explained this in the limitations section.

Additionally, you stated that "we iterated to the medical students that we wanted to hear their opinions of being a patient of a virtual PCP, rather than their opinions from a provider perspective". I personally believe that this is quite unrealistic. In fact, the qualitative approach used to address the topic is based on subjective point of view and it sounds hard to believe that the subjects involved were able to split their opinion as patient from the one as provider.

Response: This statement was intended to clarify that we wanted the students to consider using a vPCP as a patient, rather than using it as a physician in clinical practice. It was not intended as a request that they separate their educational background from their opinion. This has been clarified in the methods section.

Introduction:

You present references talking about Artificial Intelligence and telemedicine. These are completely different digital tools, please clarify and focus on the specific topic.

Response: A virtual primary care physician (vPCP) uses both AI and telehealth technologies. This has been clarified in lines 44 and 46.

Line 36: Please, provide a reference to support this statement

Response: We have added a reference here and reformed the in-text citations and reference list accordingly.

Line 37: Please, provide a reference to support this statement

Response: We have added a reference here and reformed the in-text citations and reference list accordingly.

Line 53: Please, provide a definition of Virtual Primary Care.

Response: A definition of virtual primary care has been provided here, in the third paragraph of the Introduction we have added a sub-heading, “Virtual Primary Care,” as we also define other related terms, such as remote monitoring, distinguishing them from virtual primary care.

Line 54-55: "While it is clear that technology may transform healthcare, less is known about the perspectives of its potential users". Please discuss more in depth this interesting issue (how healthcare can be transformed buy technology?) and provide more references.

Response: We have expanded this section to include recent health technology innovations, such as remote monitoring and AI-assisted radiology.

Lines 58-61: This point should be discussed more in depth, particularly focusing on how technology can improve this condition.

Response: We have expanded this section to detail the health consequences of young people not seeking healthcare, and explained how technology (specifically telehealth) may improve access and utilization among young adults.

Methods:

- How the subjects were recruited? They were selected randomly from the student list? Was the project presented in the schools and interested subjects contacted the researcher? Please provide further information on this crucial issue.

Response: Participants were recruited via mass email to departments in targeted disciplines. This information has been added to the manuscript in the 1st paragraph of the Methods section.

- "One researcher facilitated the focus groups, while two others were note-takers". The researcher was specifically trained to facilitate focus group? Some techniques were used to avoid the researcher to influence the respondents?

Response: The same researcher (author JA), who has expertise in conducting focus groups, facilitated each. A script was followed in order to avoid influence of the respondents by the facilitator. This information has been added to Data Collection paragraph in the Methods section.

- The facilitating researcher was the same in the different focus group? If no, how ensure uniformity?

Response: The same researcher facilitated each focus group. This has been clarified in the Data Collection paragraph within the Methods section.

- How the questions were developed?

Response: The questions were developed in discussions among all authors, informed by previous experience with focus groups and existing literature. This has been described in the Data Collection paragraph within the Methods section.

Discussion:

The discussion section sounds a repetition of the results section. Please provide more in depth discussion of the results.

Response: We appreciate this feedback. The discussions section has been expanded to include a more in-depth discussion of the results, specifically in relation to care disparities and ethical considerations.

References

1. Carminati L. Generalizability in qualitative research: a tale of two traditions. Qual Health Res. 2018 Nov;28(13):2094-101. doi: 10.1177/1049732318788379

2. Firestone WA. Alternative arguments for generalizing from data as applied to qualitative research. Educational Researcher. 1993 May;22(4):16-23. doi: 10.3102/0013189X022004016

3. Smith B. Generalizability in qualitative research: misunderstandings, opportunities and recommendations for the sport and exercise sciences. Qual Res Sport Exerc. 2018 Jan;10(1):137-49. doi: 10.1080/2159676X.2017.1393221

---

## [Editor Report · Decision Letter 1]

25 Nov 2020

Perceptions of virtual primary care physicians: a focus group study of medical and data science graduate students

PONE-D-20-10736R1

Dear Dr. Goetz,

We’re pleased to inform you that your manuscript has been judged scientifically suitable for publication and will be formally accepted for publication once it meets all outstanding technical requirements.

Kind regards,

Maria Rosaria Gualano, MD, MPH

Academic Editor

PLOS ONE
---

## [Editor Report · Acceptance letter]

1 Dec 2020

PONE-D-20-10736R1 

Perceptions of virtual primary care physicians: a focus group study of medical and data science graduate students 

Dear Dr. Goetz:

I'm pleased to inform you that your manuscript has been deemed suitable for publication in PLOS ONE. Congratulations! Your manuscript is now with our production department. 

Kind regards, 

on behalf of

Dr. Maria Rosaria Gualano 

Academic Editor

PLOS ONE